# The β-alanine transporter BalaT is required for visual neurotransmission in *Drosophila*

**Yongchao Han[1,2†], Liangyao Xiong[2,3†], Ying Xu[2,4], Tian Tian[2], Tao Wang[1,2*]**

[1]Graduate School of Peking Union Medical College, Chinese Academy of Medical Sciences, Beijing, China; [2]National Institute of Biological Sciences, Beijing, China; [3]Peking University-Tsinghua University-National Institute of Biological Sciences Joint Graduate Program, School of Life Sciences, Peking University, Beijing, China; [4]School of Life Sciences, Beijing Normal University, Beijing, China

**Abstract** The recycling of neurotransmitters is essential for sustained synaptic transmission. In Drosophila, histamine recycling is required for visual synaptic transmission. Synaptic histamine is rapidly taken up by laminar glia, and is converted to carcinine. After delivered back to photoreceptors, carcinine is hydrolyzed to release histamine and β-alanine. This histamine is repackaged into synaptic vesicles, but it is unclear how the β-alanine is returned to the laminar glial cells. Here, we identified a new β-alanine transporter, which we named BalaT (Beta-<u>ala</u>nine <u>T</u>ransporter). Null balat mutants exhibited lower levels of β-alanine, as well as less β-alanine accumulation in the retina. Moreover, BalaT is expressed and required in retinal pigment cells for maintaining visual synaptic transmission and phototaxis behavior. These results provide the first genetic evidence that retinal pigment cells play a critical role in visual neurotransmission, and suggest that a BalaT-dependent β-alanine trafficking pathway is required for histamine homeostasis and visual neurotransmission.

**\*For correspondence:**
wangtao1006@nibs.ac.cn

[†]These authors contributed equally to this work

**Competing interests:** The authors declare that no competing interests exist.

## Introduction

Efficient clearance of neurotransmitters from the synaptic cleft is crucial for terminating a synaptic signal. These cleared neurotransmitters must then be recycled for sustained synaptic transmission (i.e., to maintain stable levels of transmitters) (*Blakely and Edwards, 2012*). Neurons that utilize histamine as a neurotransmitter play important roles in a variety of biological processes, including cognition, sleep, and metabolism, and dysfunction of the histaminergic system in the brain has been linked to multiple neurological diseases (*Haas et al., 2008*; *Panula and Nuutinen, 2013*). Histamine is particularly important in the *Drosophila* visual system (*Hardie, 1989*), and histamine recycling is the dominant pathway for maintaining adequate levels of histamine in adult photoreceptors (*Borycz et al., 2002*). In this system, and in mammalian systems as well, the histamine recycling process involves both neurons and glial cells that surround the synapse (*Edwards and Meinertzhagen, 2010*; *Yoshikawa et al., 2013*). In fact, a network of glia cells and pigment cells interconnected via gap junctions plays a key role in the long-distance recycling of histamine in the *Drosophila* visual system (*Chaturvedi et al., 2014*). However, the mechanisms by which this cellular network recycles histamine and its metabolites are unknown.

Once histamine is released by a photoreceptor into the optic lamina, it is quickly removed from the synaptic cleft by epithelia glial cells that surround the synapse. In these laminar glia, histamine is conjugated to β-alanine by an N-β-alanyl-dopamine synthase, called Ebony, to form the inactive metabolite, carcinine (*Borycz et al., 2002*; *Richardt et al., 2003, 2002*). Recently, a carcinine

**eLife digest** Neurons transmit information around the body in the form of electrical signals, but these signals cannot cross the gaps between neurons. To send a message to its neighbor, a neuron releases a molecule known as a neurotransmitter into the gap between the cells. The neurotransmitter binds to proteins on the recipient neuron and triggers new electrical signals inside that cell. When the message has been received, the neurotransmitter molecules are returned to the first neuron so that they can be reused.

This recycling is particularly important in the visual system, where neurons communicate via rapid-fire signaling. In fruit flies, for example, light-sensitive neurons in the eye known as photoreceptors release a neurotransmitter called histamine when they detect light. Supporting cells called laminar glia take up any leftover histamine and combine it with another molecule known as $\beta$-alanine to form a larger molecule. The photoreceptors absorb this larger molecule and break it back down into histamine and $\beta$-alanine. However, it is not clear how the $\beta$-alanine returns to the glia to allow this cycle to continue.

Many molecules rely on so-called "transporter" proteins to help them move into or out of cells. To identify transporters that might help to move $\beta$-alanine, Han, Xiong et al. prepared a list of fruit fly genes that encode transporter proteins found inside the insect's head. Testing the resulting proteins in a cultured cell system revealed that one of them was able to transport $\beta$-alanine. This protein, named BalaT, is found in another type of support cell called retinal pigment cells. Mutant flies that cannot produce BalaT are blind because their photoreceptors have problems in transmitting information to other neurons.

Han, Xiong et al. propose that BalaT transports $\beta$-alanine from photoreceptors to retinal pigment cells, which then pass $\beta$-alanine on to the laminar glia. Follow-up studies are required to find out exactly how the laminar glia take up histamine.

transporter, CarT that is responsible for the uptake of carcinine from the synaptic cleft into photoreceptors has been identified (*Chaturvedi et al., 2016*; *Stenesen et al., 2015*; *Xu et al., 2015*). The discovery that CarT retrieves carcinine directly from the laminar synaptic cleft indicates that carcinine is not transported from laminar glia to photoreceptor cell bodies through a long-distance trafficking mechanism involving the laminar glial and pigment cell network.

Within photoreceptors, carcinine is hydrolyzed to generate histamine and $\beta$-alanine, a reaction catalyzed by an N-$\beta$-alanyl-dopamine hydrolase called Tan (*Borycz et al., 2002*; *True et al., 2005*). This regenerated histamine is then repackaged into synaptic vesicles for subsequent light-induced release, but it is unclear how $\beta$-alanine is returned to the laminar glial cells (where it is needed to inactivate histamine). It is also unclear whether $\beta$-alanine recycling is required for visual transmission. The finding that high levels of $\beta$-alanine are detected in retinal pigment cells suggested that pigment cells may be critical for the transport and storage of $\beta$-alanine in this system (*Borycz et al., 2012*; *Chaturvedi et al., 2014*). To address these questions, we sought to determine whether there was a $\beta$-alanine transporter that was required for *Drosophila* vision transmission.

Here, we identified BalaT, which is a plasma membrane transporter capable to transporting $\beta$-alanine into cells. Mutations in the *balat* gene disrupted photoreceptor synaptic transmission, phototaxis behaviors, and contents and retinal distribution of $\beta$-alanine. BalaT expression in retinal pigment cells completely rescued the defective visual transmission of *balat* mutants. Furthermore, the gap junction proteins Inx1 and Inx3 were required in retinal pigment cells for visual neurotransmission. We therefore provide evidence for a novel $\beta$-alanine recycling pathway, in which $\beta$-alanine that is released from photoreceptor cells is taken up and stored by retinal pigment cells, and then delivered to laminar glia via gap junctions, where histamine is inactivated by conjugation with $\beta$-alanine.

# Results

## CG3790 transports β-alanine in vitro

In *Drosophila* photoreceptors, carcinine is hydrolyzed to form histamine and β-alanine (*True et al., 2005*; *Wagner et al., 2007*). Histamine is then released as a neurotransmitter, taken up by laminar glial cells, and then conjugated with β-alanine to form carcinine. Although it is known that CarT transports carcinine into photoreceptors, it is unclear how β-alanine is delivered to laminar glia. To begin to understand this process, we sought to identify a β-alanine transporter in the fly visual system. We speculated that a transporter responsible for β-alanine uptake would likely be enriched in fly heads, so we examined previous RNA-seq data comparing mRNAs isolated from the head with mRNAs isolated from the body (*Xu and Wang, 2016*). Among ~600 putative transmembrane transporters encoded by the *Drosophila* genome, we identified 20 head-enriched genes (*Table 1*) that represented potential β-alanine transporters (*Figure 1A*).

To determine if any of these candidate transporters could transport β-alanine, we used HEK293T cells to perform uptake assays. Each candidate was transiently expressed in HEK293T cell, and their ability to uptake [³H]-β-alanine was assessed. Only one candidate transporters, CG3790, exhibited β-alanine uptake activity (*Figure 1A*). The β-alanine content of CG3790-transfected cells was

**Table 1.** Description of 20 head-enriched genes.

| CG number | Name | Description |
| --- | --- | --- |
| CG3790 | | Organic anion transmembrane transporter activity |
| CG18660 | Nckx30C | Sodium/potassium/calcium exchanger |
| CG2893 | zydeco | Potassium-dependent sodium/calcium exchanger |
| CG7342 | | Organic anion transmembrane transporter activity |
| CG13610 | Orct2 | Organic cation transporter |
| CG6331 | Orct | Organic anion transmembrane transporter activity |
| CG6126 | | Organic anion transmembrane transporter activity |
| CG4360 | | Nucleic acid binding |
| CG3159 | Eaat2 | L-aspartate transmembrane transporter activity |
| CG6723 | | Sodium/solute symporter |
| CG11010 | Ent3 | Nucleoside transmembrane transporter activity |
| CG13743 | | Amino acid transporter |
| CG10804 | | Neurotransmitter transporter activity |
| CG4545 | SerT | Monoamine transmembrane transporter activity |
| CG5281 | | EamA domain |
| CG43066 | | Neurotransmitter transporter activity |
| CG7442 | SLC22A | Choline transmembrane transporter activity |
| CG7708 | | Sodium/solute symporter |
| CG13248 | | Amino acid/polyamine transporter I |
| CG42322 | | Solute carrier family 35 member F3/F4 |

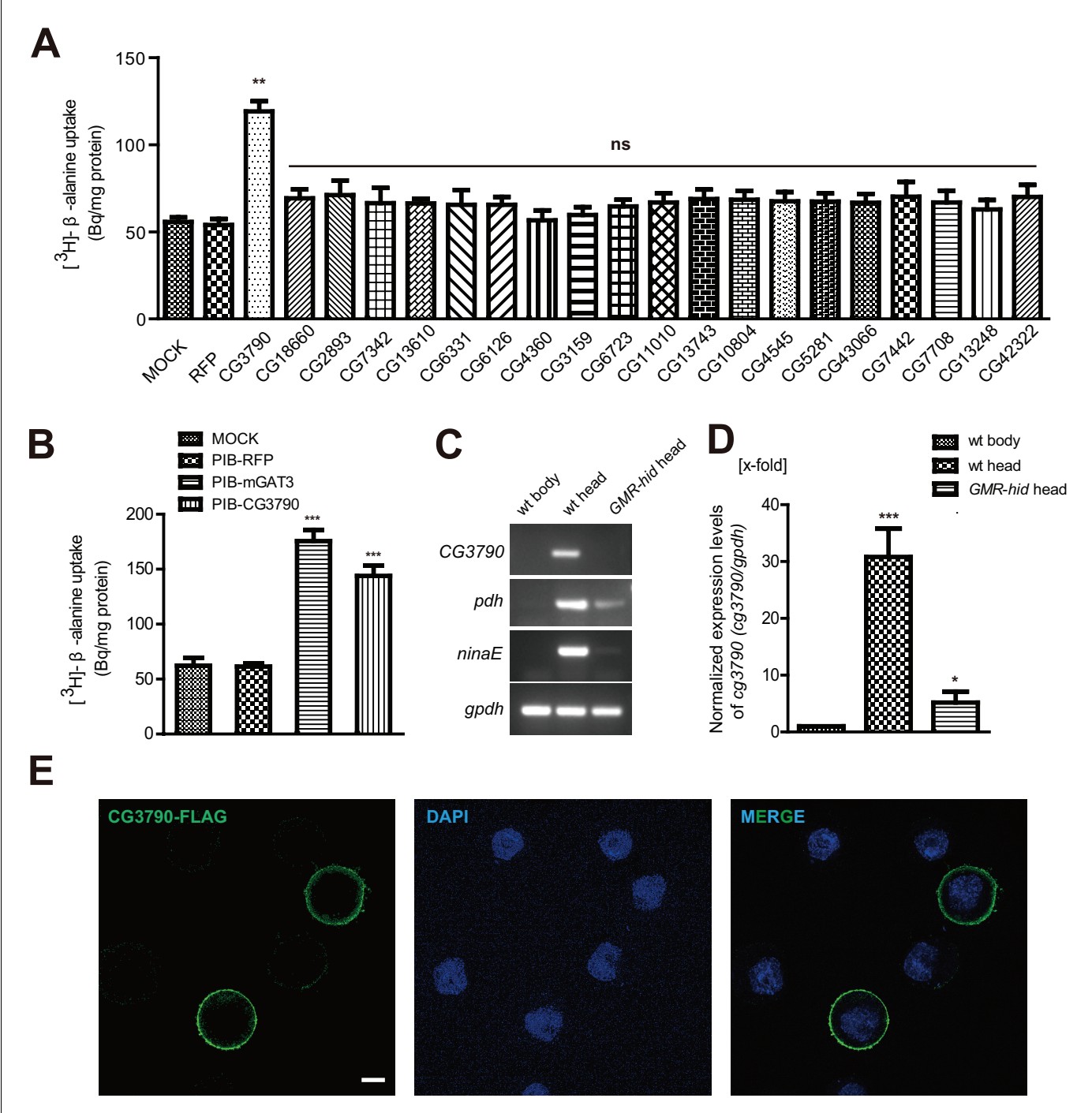

**Figure 1.** CG3790 is a retina-enriched $\beta$-alanine transporter. (**A**) Screening for a $\beta$-alanine transporter. HEK293T cells transiently expressing one of ~20 head-enriched potential transporters were exposed to [$^3$H]-$\beta$-alanine, which was added to the ECF buffer (final concentration $3.7 \times 10^4$ Bq). RFP (red fluorescent protein) was expressed as a negative control. (**B**) CG3790 transported $\beta$-alanine into S2 cells. Mouse GAT3 and RFP were used as positive and negative control, respectively. The results given are the mean values ± S.D. of four experiments. (**C–D**) CG3790 was expressed at high levels in the compound eye. (**C**) Relative RNA transcript levels (RT-PCR experiments) show that CG3790 expression is enriched in wild-type ($w^{1118}$) heads compared with wild-type bodies or *GMR-hid* heads. (**D**) Relative *CG3790* transcript levels from wild-type (wt) bodies, wt heads, and *GMR-hid* heads (*gpdh* served as an internal control). RNA levels were normalized to levels in wt bodies, which were set to 1. Error bars indicate standard deviations (SDs) from three replicate experiments. Significant differences between candidates and control were determined using unpaired t-tests (**p<0.01; ns, not significant). (**E**) S2 cells were transiently transfected with 3xFlag-tagged CG3790, and then labeled with Flag antibody (green) and DAPI (blue). Scale bar, 2 µm.

*Figure 1 continued on next page*

*Figure 1 continued*

The following source data and figure supplements are available for figure 1:

**Source data 1.** Relates to *Figure 1A and B*.
**Source data 2.** Relates to *Figure 1D*.
**Figure supplement 1.** CG3790 is a SLC22 family protein.
**Figure supplement 2.** CG3790 is unable to transport histamine and GABA.
**Figure supplement 2—source data 1.** Relates to *Figure 1—figure supplement 2A and B*.
**Figure supplement 3.** *GMR-hid* abolished expression of *ninaE* and *pdh*.
**Figure supplement 3—source data 1.** Relates to *Figure 1—figure supplement 3A and B*.

approximately 119 Bq/mg, which was 2.7-fold greater than measured for mock- or RFP-transfected controls (55 and 53 Bq/mg, respectively). CG3790 shares significant amino acid identity with the mammalian solute carrier family 22 (SLC22), which includes both mouse and human OCT3 (*Figure 1—figure supplement 1*). The $\beta$-alanine transporting activity of CG3790 was further confirmed in S2 cells (*Figure 1B*). Mouse GAT3 (GABA transporter type 3), which is known to efficiently take up $\beta$-alanine, exhibited levels of $\beta$-alanine transporting activity that were similar to CG3790, suggesting that CG3790 is a *bona fide* $\beta$-alanine transporter (*Christiansen et al., 2007*) (*Figure 1B*). We next sought to determine whether CG3790 is a specific $\beta$-alanine transporter in *Drosophila*. Because a histamine transporter has not yet been identified, we first asked whether CG3790 is able to transport histamine. Histamine uptake assays revealed that CG3790 does not exhibit histamine uptake activity. As a control, the human Organic Cation Transporter (OCT2), which is known to take up histamine, exhibited high levels of histamine transport activity when expressed in S2 cells (*Figure 1—figure supplement 2A*). Since mGAT3 is also a GABA transporter, we asked whether CG3790 is able to transport GABA. While mGAT3 efficiently transported GABA, CG3790 did not exhibit GABA transporting activity when expressed in S2 cells (*Figure 1—figure supplement 2B*). Moreover, when we expressed Flag-tagged CG3790 in S2 cells the Flag signal localized exclusively to the plasma membrane (*Figure 1E*). These data suggest that *CG3790* encodes a plasma membrane $\beta$-alanine transporter.

To further confirm that the CG3790 transporter functions in the fly head, we performed quantitative PCR comparing mRNAs isolated from wild-type heads, wild-type bodies, or heads from *GMR-hid* flies, which are devoid of eyes. As is seen with other eye-specific genes including *ninaE* (*neither inactivation nor afterpotential E*), which is expressed exclusively in photoreceptor cells (*O'Tousa et al., 1985*), and *pdh* (*pigment-cell-enriched dehydrogenase*), which is expressed exclusively in retinal pigment cells) (*Wang et al., 2010*), *CG3790* mRNA transcripts were absent from fly bodies (*Figure 1C–D* and *Figure 1—figure supplement 3A–B*). Ectopic expression of the pre-apoptotic gene *hid* via the *glass multiple reporter* (*GMR*) promoter results in eye ablation (*Grether et al., 1995*; *Hay et al., 1994*). As such, *ninaE* and *pdh* expression is greatly reduced in the heads of *GMR-hid* flies. Similarly, *CG3790* transcript levels were also greatly reduced in the heads of *GMR-hid* flies, suggesting that *CG3790* is primarily expressed in the fly compound eye (*Figure 1C–D* and *Figure 1—figure supplement 2A–B*). These findings indicate that *CG3790* encodes a retina-enriched plasma membrane $\beta$-alanine transporter. We therefore named this gene *balat* (beta-alanine transporter).

## BalaT is required for visual synaptic transmission

To study the function of BalaT in visual perception, we generated two null mutations in the *balat* gene by deleting ~900 bp and ~660 bp genomic fragments using the CRISPR-associated single-guide RNA system (Cas9) (*Figure 2A*). PCR amplification of the *balat* locus from genomic DNA

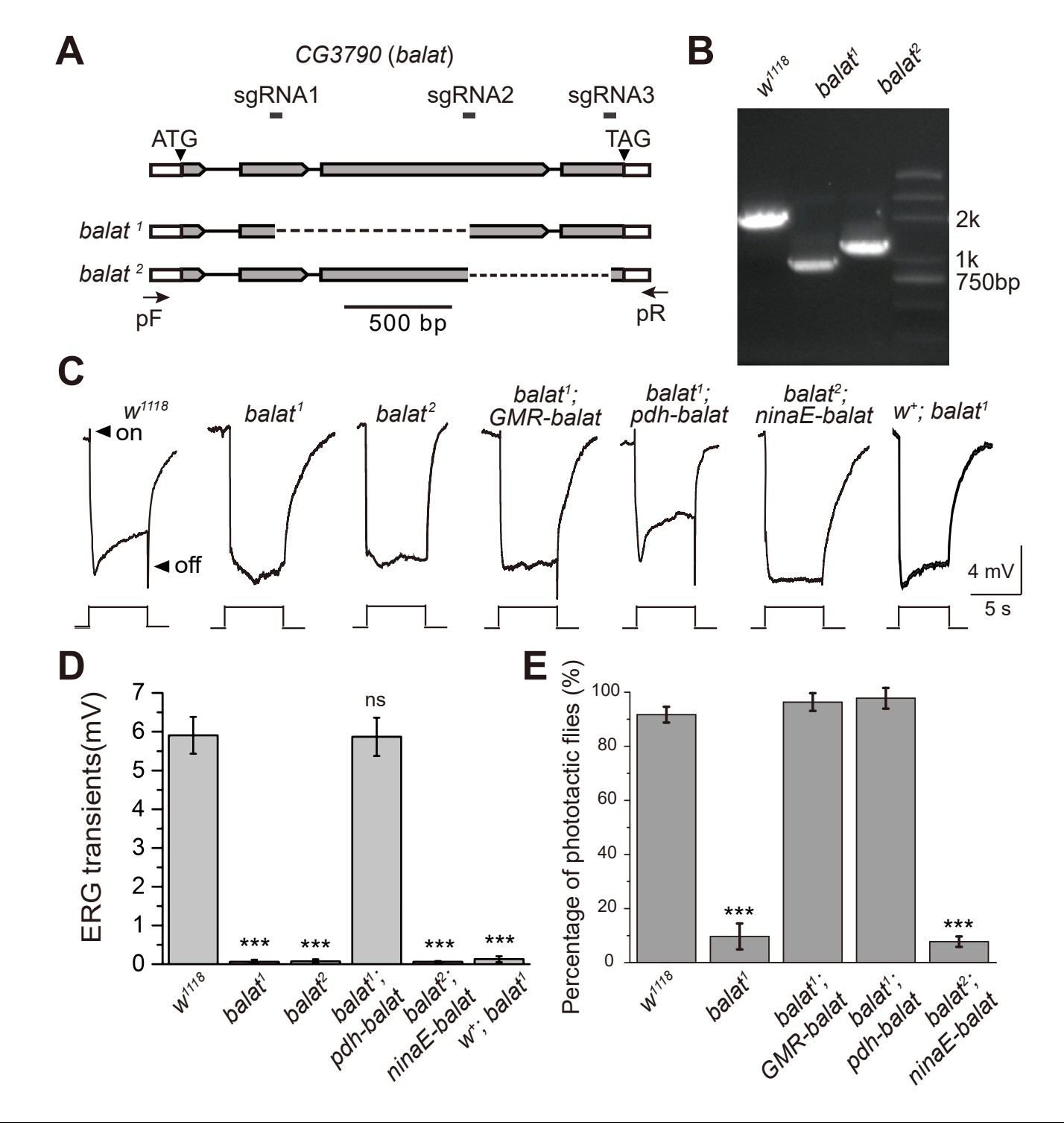

**Figure 2.** Mutations in *balat* disrupt photoreceptor cell synaptic transmission and visual behavior. (**A**) Schematic for *balat* knock-out by sgRNA targeting. The organization of the *balat* locus and the expected structures of *balta¹* and *balat²* alleles are shown. Boxes represent exons with the coding region between ATG and TAG. The sgRNA1 and sgRNA2 primer pair was used to generate the *balat¹* allele; the sgRNA2 and sgRNA3 primer pair was used to generate the *balat²* allele. Arrows indicate the primers used for genomic PCR. (**B**) PCR products obtained from *balat¹* and *balat²* mutants show successful gene deletions. (**C**) ERG recordings from: wild-type (*w¹¹¹⁸*), *balat¹*, *balat²*, *balat¹;GMR-balaT*, *balat¹;pdh-balaT*, *balat²;ninaE-balaT* and *w⁺; balat¹* flies. Young flies (<3 days after eclosion) were dark adapted for 1 min and subsequently exposed to a 5 s pulse of orange light. ON and OFF transients are indicated by arrows. All flies contained the *w¹¹¹⁸* mutation except for the *w⁺; balat¹* flies. (**D**) Quantitative analysis of the
*Figure 2 continued on next page*

Figure 2 continued

amplitudes of ERG OFF transients shown in C. (E) Phototaxis behaviors of wt, *balat¹ balat¹;GMR-balaT*, *balat¹;pdh-balaT* and *balat²;ninaE-balaT* flies. Significant differences between mutant and wild-type flies were determined using unpaired t-tests (***p<0.001; ns, not significant).

The following source data and figure supplement are available for figure 2:

**Source data 1.** Relates to *Figure 2D and E*.
**Figure supplement 1.** Quantification of the ERG ON transients.

isolated from wild-type, *balat¹*, and *balat²* flies revealed a truncated *balat* locus in the mutant samples. Thus, the *balat* locus is disrupted in *balat¹* and *balat²* flies (*Figure 2B*).

These *balat* mutations did not cause lethality or other visible phenotypes, so we performed electroretinogram (ERG) recordings to determine whether BalaT functions in phototransduction. Exposing wild-type flies to light results in two primary components in the ERG recording, including a sustained corneal negative response resulting from photoreceptor depolarization, and 'on' and 'off' transients resulting from postsynaptic neuronal activity in the optic lamina (downstream of the photoreceptors) (*Wang and Montell, 2007*) (*Figure 2C*). Mutations in genes that disrupt histamine recycling exhibit reduced 'on' and 'off' transients, reflecting defective synaptic transmission of photoreceptors (*Xu et al., 2015*). Similarly, both *balat¹ and balat²* mutant flies lacked ERG 'on' and 'off' transients, indicating that BalaT is required for visual synaptic transmission (*Figure 2C and D*, and *Figure 2—figure supplement 1*). To confirm that the loss of ERG transients resulted from mutation of the *balat* locus, we generated a *GMR-balat* transgenic fly, which expresses BalaT in the compound eyes under control of the *GMR* promoter. The *GMR-balat* transgene completely rescued the 'on' and 'off' transients when crossed into the *balat¹* mutant flies (*Figure 2C*). Mutations in the *white* gene, which encodes an ATP-binding cassette (ABC) transporter, has been reported to affect histamine levels (*Borycz et al., 2008*). To make sure that the *white* mutation was not affecting ERG phenotypes exhibited by *balat* mutants, we crossed the *balat* mutation into a wild-type *white* ($w^+$) background and found that 'on' and 'off' transients were still absent in this $w^+$; *balat¹* fly (*Figure 2C and D*).

Disrupting visual transmission results in blindness, which is reflected in the loss of phototaxis behavior (*Behnia and Desplan, 2015*). We next used this behavioral assay to assess the ability of *balat* mutant flies to see. Consistent with the loss of 'on' and 'off' transients, *balat* mutant flies exhibited impaired phototaxis, which was fully restored by the *GMR-balat* transgene (*Figure 2E*). These results strongly support the conclusion that BalaT is involved in visual transmission and that the loss of BalaT severely disrupts vision.

## BalaT localizes to pigment cells

It is not currently known how $\beta$-alanine is transported out of the photoreceptor cells once generated (along with histamine) by the Tan hydrolase. Having established that BalaT functions as a $\beta$-alanine transporter in the retina, we next sought to determine which cells express *balat*, with the goal of determining the site of $\beta$-alanine transport during phototransduction. As we were unable to generate a high affinity antibody against BalaT, we used CRISPR/Cas9-based genome editing to introduce mCherry into the *balat* locus, downstream of the native *balat* promoter (*balat-mcherry*) (*Figure 3—figure supplement 1A*). We identified *balat-mcherry* flies though PCR and RFP immunofluorescence (*Figure 3—figure supplement 1B–C*). Importantly, homozygous *balat-mcherry* flies displayed the loss of 'on' and 'off' transients, as expected, indicating the insertion of *mcherry* into the *balat* locus (*Figure 3—figure supplement 1D*).

Photoreceptor cells and retinal pigment cells are the two major cell types in the compound eye. Staining with phalloidin and PDH labels the rhabdomere regions of photoreceptor cells and the retinal pigment cells, respectively. The mCherry signals in the *balat-mcherry* retina were predominantly detected in retinal pigment cells, as it overlapped with the PDH signal (*Figure 3A and B*). No mCherry signaling was detected in wild-type flies (*Figure 3C and D*).

Since BalaT localized to retinal pigment cells, we next asked whether BalaT is required in pigment cells by performing tissue-specific rescue experiments. We expressed BalaT specifically in

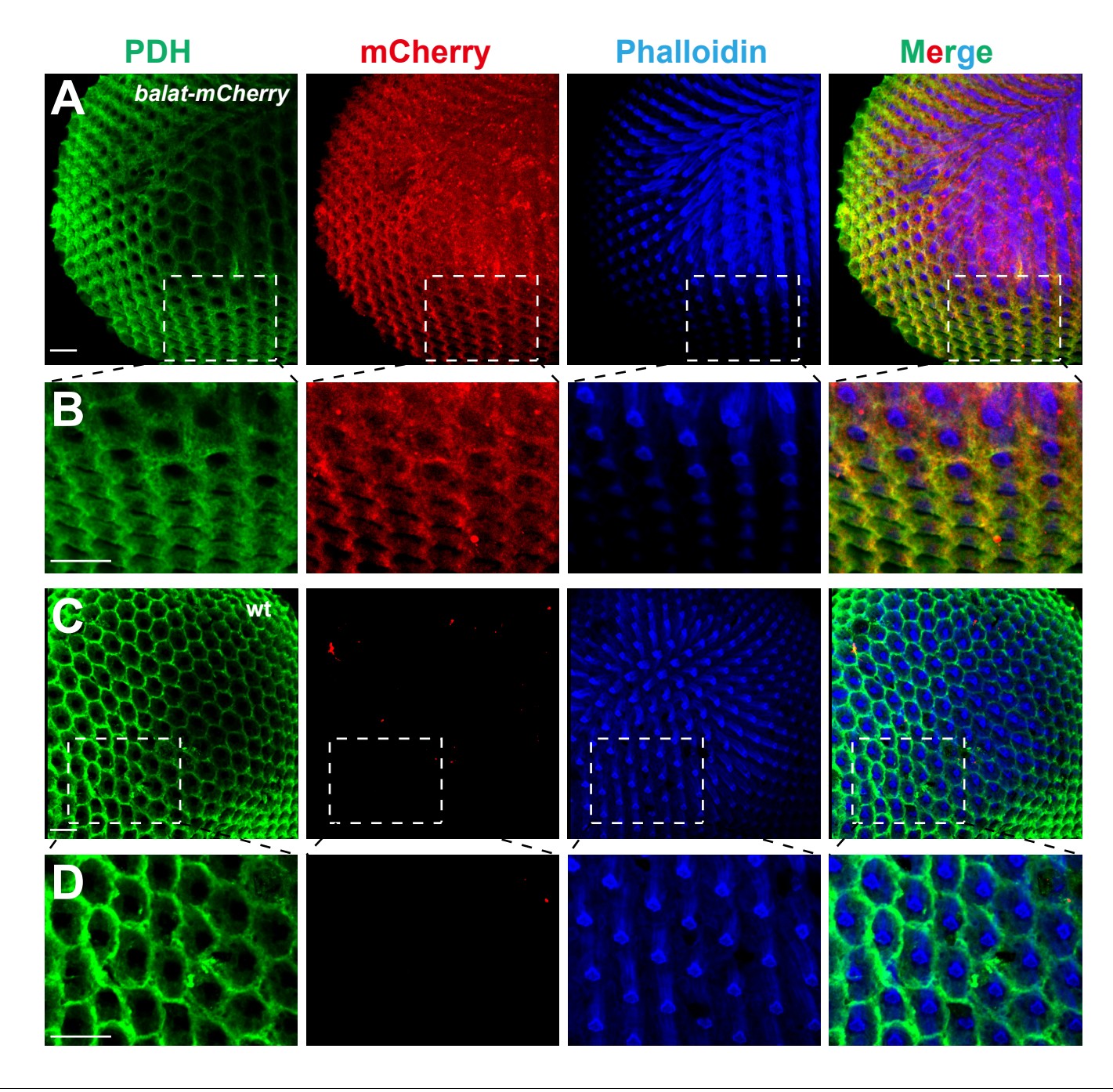

**Figure 3.** BalaT localizes to pigment cells. (A–B) Compound eyes from *balat-mcherry* flies that express mCherry (red) driven by the endogenous *balat* promoter were labeled for PDH (green), mCherry (red), and phalloidin (blue). High-magnification images are shown in (B). (C–D) mCherry signals were not detected in wild-type (wt) retinas. Scale bars represent 20 μm.

The following figure supplement is available for figure 3:

**Figure supplement 1.** Generation of *balat-mcherry* flies.

photoreceptor cells using the *ninaE* promoter (*ninaE-balat*) or in retinal pigment cells using the *pdh* promoter (*pdh-balat*). The *pdh-balat* transgene fully restored 'on' and 'off' transients and phototaxic behaviors when crossed into *balat¹* mutant flies, whereas expressing BalaT in photoreceptor cells failed to rescue these phenotypes (*Figure 2C–E*, and *Figure 2—figure supplement 1*). These results indicate that BalaT functions in pigment cells, and that pigment cells may play an important role in transporting β-alanine from photoreceptors to epithelia glial cells in the optic lamina.

## Inx1 and Inx3 are required for visual synaptic transmission

If β-alanine is transported into pigment cell by BalaT, it must then be transported from the pigment cells to the laminar glial cells, where it can be conjugated with histamine to form carcinine. It has recently been reported that a multicellular network involving laminar glia and pigment cells mediates long-distance recycling of histamine metabolites (*Chaturvedi et al., 2014*). Since this transport relies on the gap junction protein Innexin 2 (Inx2) in glia, it is possible that gap junctions between pigment cells and laminar glial cells are important for β-alanine recycling, and thus for maintaining visual synaptic transmission. We first asked whether disruption of Inx2 in the retina affects fly visual transmission. Knocking down Inx2 in the retina (*inx2^RNAi^: GMR-gal4/UAS-inx2^RNAi^*) did not affect ERG transients (*Figure 4A and C*). However, there are seven Innexins, and Inx1, Inx2, and Inx3 are all expressed in the adult head. Knockdown of Inx1 or Inx3 in the compound eye also did not disrupt the ERG 'on' and 'off' transients (*Figure 4B and D*). Because these Innexins may have redundant functions, we knocked down combinations of two retinal Innexins and measured the effect on fly visual transmission. Knockdown of Inx1/Inx2 or Inx2/Inx3 did not affect ERG transients, but 'on' and 'off' transients were largely reduce by Inx1/Inx3 double knockdown (*Figure 4E–4G*). These results indicated that the gap junction proteins Inx1 and Inx3 are required in compound eyes for visual transmission.

## β-alanine levels and distributions are altered in *balat* mutant flies

We next examined the in vivo levels of β-alanine, histamine, and carcinine in the heads of *balat* mutant flies via liquid chromatography-mass spectrometry (LC-MS). For *cart¹* mutant flies, which cannot transport carcinine into photoreceptor cells, levels of histamine and β-alanine were significantly reduced (*Figure 5A and B*), as has been reported (*Xu et al., 2015*). Similarly, levels of β-alanine and histamine were significantly reduced in *balat¹* mutant flies (*Figure 5A and B*). Reduction of histamine in *balat¹* mutant flies indicates defective histamine recycling and explains the disruption of photoreceptor synaptic transmission in *balat¹* mutants. Reductions in β-alanine may result from inadequate β-alanine storage and/or transport into pigment cells (*Borycz et al., 2012*). These results therefore support the hypothesis that BalaT transports β-alanine into retinal pigment cells.

If flies are not able to transport β-alanine into pigment cells for storage, β-alanine should be specifically decreased in the retina. To test this idea, we sectioned fly heads and labeled them with an antibody against β-alanine. In wild-type flies, levels of β-alanine were highest in the retina, consistent with previous analyses (*Borycz et al., 2012*; *Chaturvedi et al., 2014*) (*Figure 5C and F*). In contrast, levels of β-alanine were lower in *balat¹* sections (for comparison, retinal signals were normalized to the β-alanine signal in the lobula) (*Figure 5D and F*). *GMR-balat* restored retinal β-alanine levels in *balat¹* mutants (*Figure 5E and F*), and rescued photoreceptor synaptic transmission. Reduced levels of β-alanine in retina lacking BalaT support the hypothesis that BalaT transports β-alanine into retinal pigment cells. We next checked if reductions in β-alanine change its distribution. In *pyd1* and *black* double mutants (*pyd1;b¹*), two independent metabolic pathways for de novo β-alanine biosynthesis are blocked and thus β-alanine levels are largely reduced (*Borycz et al., 2012*). However, in *pyd1;b¹* flies the pattern of β-alanine distribution is the same as seen in wild type, although immune intensity is reduced, as is the case with *balat¹* mutant flies (*Figure 5—figure supplement 1*). Therefore, pigment cells may serve as a reservoir of stored β-alanine through a BalaT-dependent mechanism. Since Inx1/Inx3 double knockdown flies showed significant reductions in 'on' and 'off' transients, we asked whether Inx1/Inx3 is required for β-alanine trafficking. Of importance, β-alanine primarily accumulated in the retinal layer of Inx1/Inx3 double knock-down flies compared with control flies with either *CG2198* or *CG9962* RNAi (*Figure 4H–J*). Meanwhile in vivo levels of β-alanine in heads of Inx1/Inx3 double knockdown flies are significantly higher than in control flies (*Figure 4—figure supplement 1*). These results further support our hypothesis that Inx1/Inx3 functions to transport β-alanine back

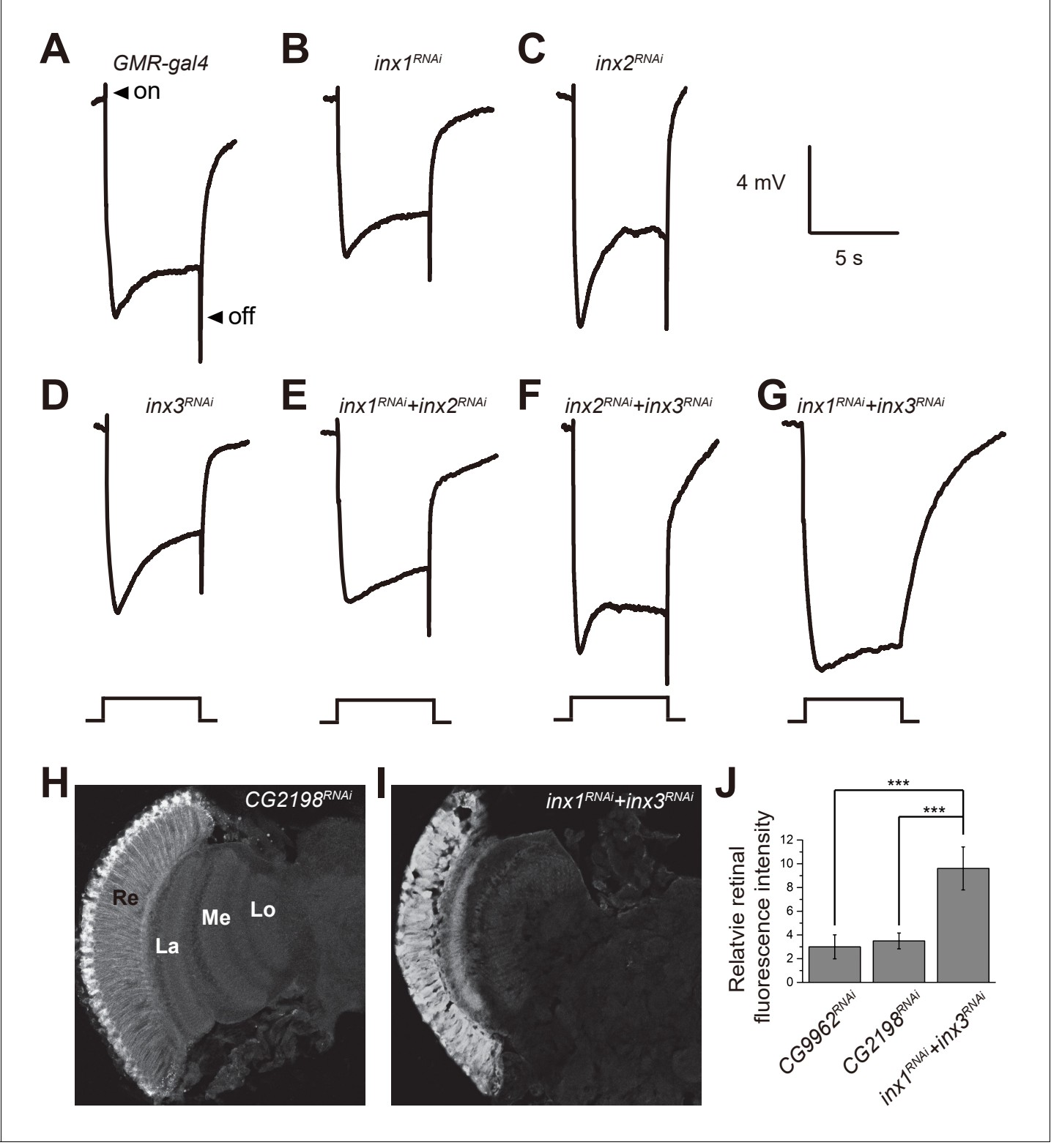

**Figure 4.** Specific knockdown of Inx1 and Inx3 in compound eyes blocks visual neurotransduction. ERG recordings of (**A**) control (*GMR-gal4*), (**B**) *inx1$^{RNAi}$* (*GMR-gal4/ UAS-inx1$^{RNAi}$*), (**C**) *inx2$^{RNAi}$* (*GMR-gal4/UAS-inx2$^{RNAi}$*), (**D**) *inx3$^{RNAi}$* (*GMR-gal4/ UAS-inx3$^{RNAi}$*), (**E**) *inx1$^{RNAi}$ + inx2$^{RNAi}$* (*GMR-gal4/ UAS-inx1$^{RNAi}$ UAS-inx1$^{RNAi}$*), (**F**) *inx2$^{RNAi}$ + inx3$^{RNAi}$* (*GMR-gal4/UAS-inx2$^{RNAi}$ UAS-inx3$^{RNAi}$*), and (**G**) *inx1$^{RNAi}$ + inx3$^{RNAi}$* (*GMR-gal4/UAS-inx1$^{RNAi}$ UAS-inx3$^{RNAi}$*) are shown. Arrows indicate ON and OFF transients in (**A**). Flies (~1 day after eclosion) were dark adapted for 1 min and subsequently exposed to a 5 s pulse of orange light. (**H–I**) β-alanine was immunolabeled in horizontal sections of heads from (**H**) control: *CG2198$^{RNAi}$* (*GMR-gal4/UAS-*
*Figure 4 continued on next page*

*Figure 4 continued*

*CG2198^{RNAi}*) and (I) *inx1^{RNAi}* + *inx3^{RNAi}* (*GMR-gal4/UAS-inx1^{RNAi} UAS-inx3^{RNAi}*) flies. Re, retina; La, lamina; Me, medulla; Lo, lobula. (J) Fluorescence intensity ratios of *β*-alanine signals between retina and lobula. Quantifications of all genotypes are averages of six replicate experiments. Significant differences between *inx1^{RNAi}* + *inx3^{RNAi}* and controls (*CG2198^{RNAi}* and *CG9962^{RNAi}*) were determined using unpaired t-tests (***$p<0.001$; ns: not significant).

The following source data and figure supplements are available for figure 4:

**Source data 1.** Relates to *Figure 4J*.
**Figure supplement 1.** Accumulation of *β*-alanine in heads of *Inx1* and *Inx3* double RNAi flies.
**Figure supplement 1—source data 1.** Relates to *Figure 4—figure supplement 1*.

into glia cell. Taken together, we have identified a previously uncharacterized *β*-alanine transporter, which we have named BalaT, and shown that BalaT is responsible for *β*-alanine transportation and storage in pigment cells.

## Discussion

Histamine is the primary neurotransmitter released by photoreceptors in flies, and the synthesis of histamine can occur de novo by photoreceptor cell-specific histidine decarboxylase (*Burg et al., 1993*; *Hardie, 1987*). However, during light stimulation, flies depend primarily on histamine recycling to sustain tonic visual transmission. Histamine is inactivated when Ebony catalyzes its conjugation to *β*-alanine in laminar glial cells (*Borycz et al., 2002*; *Richardt et al., 2002*). In photoreceptors, CarT and Tan are involved in transporting carcinine and regenerating histamine, respectively (*Chaturvedi et al., 2016*; *Stenesen et al., 2015*; *Wagner et al., 2007*; *Xu et al., 2015*). Histamine is then loaded into synaptic vesicles, but fate of the *β*-alanine generated by Tan remains unknown. Here, we identify a new *β*-alanine transporter, BalaT, and provide evidence that BalaT is required for visual transmission. BalaT is a plasma membrane protein, and is sufficient to facilitate *β*-alanine uptake in cultured cells. BalaT is expressed predominantly by retinal pigment cells, and the specific expression of BalaT in these cells rescues photoreceptor synaptic transmission in *balat^1* mutant flies. The reduced levels and altered distribution of *β*-alanine in the heads of *balat^1* mutants support the hypothesis that BalaT functions to transport *β*-alanine. Therefore, we suggest that a novel *β*-alanine transporter, BalaT, functions in retinal pigment cells to help recycle *β*-alanine.

Disrupting the synthesis of histamine and carcinine abolishes photoreceptor synaptic transmission (*Borycz et al., 2002*; *Burg et al., 1993*). The transport of *β*-alanine seems less important, as it is present at higher concentrations in Drosophila. Two common metabolic pathways have been shown to synthesize *β*-alanine: (1) aspartate decarboxylase, encoded by the *black* gene (*Hodgetts, 1972*; *Phillips et al., 2005*), and (2) the uracil degradation pathway, which includes the enzymes dihydropyrimidine dehydrogenase (Pyd1), dihydropyrimidine amidohydrolase (Pyd2), and *β*-ureidopropionase (Pyd3) (*Piskur et al., 2007*; *Rawls, 2006*). The Black protein is enriched in lamina glia, together with the protein Ebony, and functions as the major enzyme in generating *β*-alanine (*Ziegler et al., 2013*). Although levels of *β*-alanine are reduced in the heads of *black* mutants (or *pyd1* and *black* double mutants) compared to wild type, these flies still exhibit ERG transients (*Borycz et al., 2012*; *Ziegler et al., 2013*). Therefore, an efficient *β*-alanine recycling systems has been proposed to compensate for the rapid loss of *β*-alanine that occurs in the conversion of histamine to carcinine (*Borycz et al., 2012*). Here we show that mutations in *balat* disrupted transport and storage of *β*-alanine in the retina, resulting in dramatically reduced photoreceptor synaptic activity and visual behavior. Thus, we propose that Balat plays a key role in shuttling *β*-alanine from photoreceptor cells to glial cells.

A similar recycling system has been demonstrated for histamine. Mutations in *Hdc* (histidine decarboxylase) result in the loss of visual transduction. Feeding these flies histamine fully restores the visual transduction, whereas supplying histamine fails to rescue the ERG phenotypes of *ebony* mutants or *Hdc;ebony* double mutant flies (*Melzig et al., 1996*, *1998*; *Ziegler et al., 2013*). In

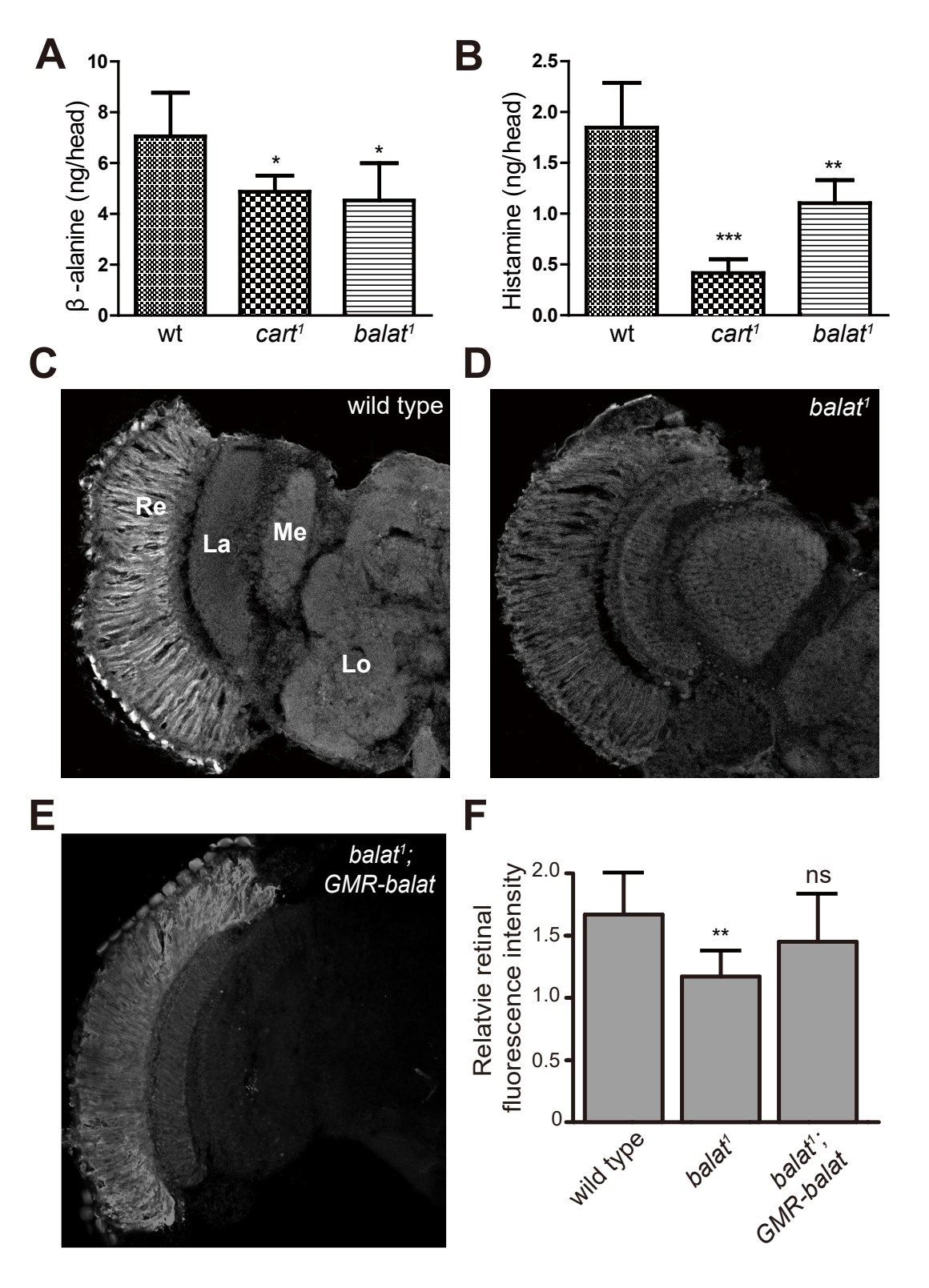

**Figure 5.** Loss of BalaT affects the level and distribution of $\beta$-alanine in vivo. (**A–B**) Levels of (**A**) $\beta$-alanine and (**B**) histamine in heads of wild-type (wt, $w^{1118}$), $cart^1$, and $balat^1$ flies are shown. Error bars indicate SDs, and significant differences between mutant and wt flies were determined using unpaired t-tests (*p<0.05; **p<0.01). (**C–E**) $\beta$-alanine was immunolabeled in horizontal sections of heads from (**C**) wild-type ($w^{1118}$), (**D**) $balat^1$, and (**E**)
*Figure 5 continued on next page*

*Figure 5 continued*

*balat[1];GMR-balat* flies. Re, retina; La, lamina; Me, medulla; Lo, lobula. (**F**) Fluorescence intensity ratios of *β*-alanine labeling between retina and lobula.
Significant differences between mutant and wild-type flies were determined using unpaired t-tests (**p<0.01; ns: not significant).

The following source data and figure supplement are available for figure 5:

**Source data 1.** Relates to *Figure 5A and B*.
**Figure supplement 1.** Distribution of *β*-alanine was unaffected in *pyd1* and *balck* double mutant flies.

*pyd1;black* double mutants, *β*–alanine levels are still higher than histamine, which may be because of the uptake of exogenous *β*-alanine (*Borycz et al., 2012*). Meanwhile, the *β*-alanine recycling pathway is able to sustain visual synaptic transmission in *black* mutant flies, even in the absence of in situ *β*-alanine generation. Therefore, *β*-alanine recycling is essential for maintaining histamine homeostasis and photoreceptor synaptic transmission. In addition, establishing and maintaining a *β*-alanine pool play key roles in sustaining visual synaptic transmission.

The majority of pigment granules, which reduce the intensity of light exposure, are formed in retinal pigment cells of the *Drosophila* compound eye. Recently, it has been demonstrated that retinal pigment cells are functionally equivalent to cells within the mammalian retinal pigment epithelium (RPE), in terms of chromophore cycling (*Travis et al., 2007*; *Wang et al., 2010*, *2012*). Just like chromophore (a subunit of rhodopsin) is sent back to pigment cells for regeneration, it has been suggested that the metabolism of *β*-alanine involves pigment cells (*Borycz et al., 2012*). Several observations indicate that BalaT functions in retina pigment cells to help regenerate histamine. Most notably, BalaT is expressed in pigment cells and visual transmission is rescued in *balat* mutants by the expression of wild-type BalaT in pigment cells, but not photoreceptor neurons. Our work provides strong genetic evidence for the involvement of retinal pigment cells in histamine recycling, and suggests that BalaT is the key player responsible for the transport of *β*-alanine into pigment cell.

It has been suggested that the transportation of histamine metabolites within the *Drosophila* visual system is mediated by a network of retinal pigment cells and laminar glia that are interconnected by gap junctions, and that this transport network is essential for fly vision (*Chaturvedi et al., 2014*). However, histaminergic photoreceptor cells directly acquire carcinine from synaptic terminals within the optic lamina through a CarT-dependent trafficking pathway (*Xu et al., 2015*). Here we provided evidence that the intercellular gap junctions between retinal pigment cells and laminar glial cells are involved in trafficking *β*-alanine, another histamine metabolite. Although the Inx2 gap junction protein is required in laminar glial cells for visual transmission, Inx2 was not required in retinal pigment cells. In contrast, downregulating the gap junction proteins Inx1 and Inx3 within the compound eye greatly impaired photoreceptor synaptic transmission. These data indicate that the Inx1 and Inx3 proteins are required in retinal pigment cells for maintaining fly vision. It has also been suggested that gap junctions may play a role in the formation of synapses between the retina and lamina in the *Drosophila* visual system (*Curtin et al., 2002*). Therefore, an alternative explanation would be that Inx1 and Inx3 are required for the development of synapses between retinal and laminar neurons. Although electron microscopy studies in *Drosophila* have not revealed gap junctions between retinal pigment cells and laminar fenestrated glia, accumulation of *β*-alanine in the retinal layer of *inx1/inx3* double mutant flies indicates defective *β*-alanine trafficking.

BalaT belongs to the SLC22 family of transporters, which mediate sodium-independent transport of neurotransmitters, amino acids, and energy metabolites (*Koepsell, 2013*). We provide evidence that BalaT functions specifically in pigment cells as a *β*-alanine transporter, as BalaT mediated *β*-alanine transport in vitro, and is expressed predominately in retinal pigment cells. The reduced levels and altered distribution of *β*-alanine in the heads of *balat* null mutant flies supports this hypothesis. *β*-alanine is one of the two constituents of the naturally occurring dipeptides, carnosine and carcinine, and is considered rate-limiting for their synthesis. Both carnosine and carcinine are found in the retina where they exert protective effects. This is because they function as antioxidants, scavenging toxic activated oxygen species (*Marchette et al., 2012*; *Pfister et al., 2011*). There is evidence that several members of the SLC family of proteins that can transport*β*-alanine are present in the retina,

including taurine transporter (Slc6a6/TauT) (*Liu et al., 1993*). This suggests a conserved function for β-alanine trafficking in mammals and *Drosophila*.

Our current work, together with previous reports, provides evidence for the formulation of a more complete histamine recycling pathway, which is critical for sustaining photoreceptor synaptic transmission (*Figure 6*). In this pathway, carcinine is synthesized in epithelial glial cells within the optic lamina, and is transported back to photoreceptor cells via the synaptic cleft. Once in the photoreceptor cells it is hydrolyzed into histamine and β-alanine by Tan, and histamine is used as neurotransmitter. Meanwhile, β-alanine is released via an unknown mechanism, and is subsequently delivered to and stored within pigment cells by BalaT. Finally, β-alanine is transported back to epithelial glia through the network of retinal pigment cells and laminar glia for histamine inactivation and carcinine synthesis.

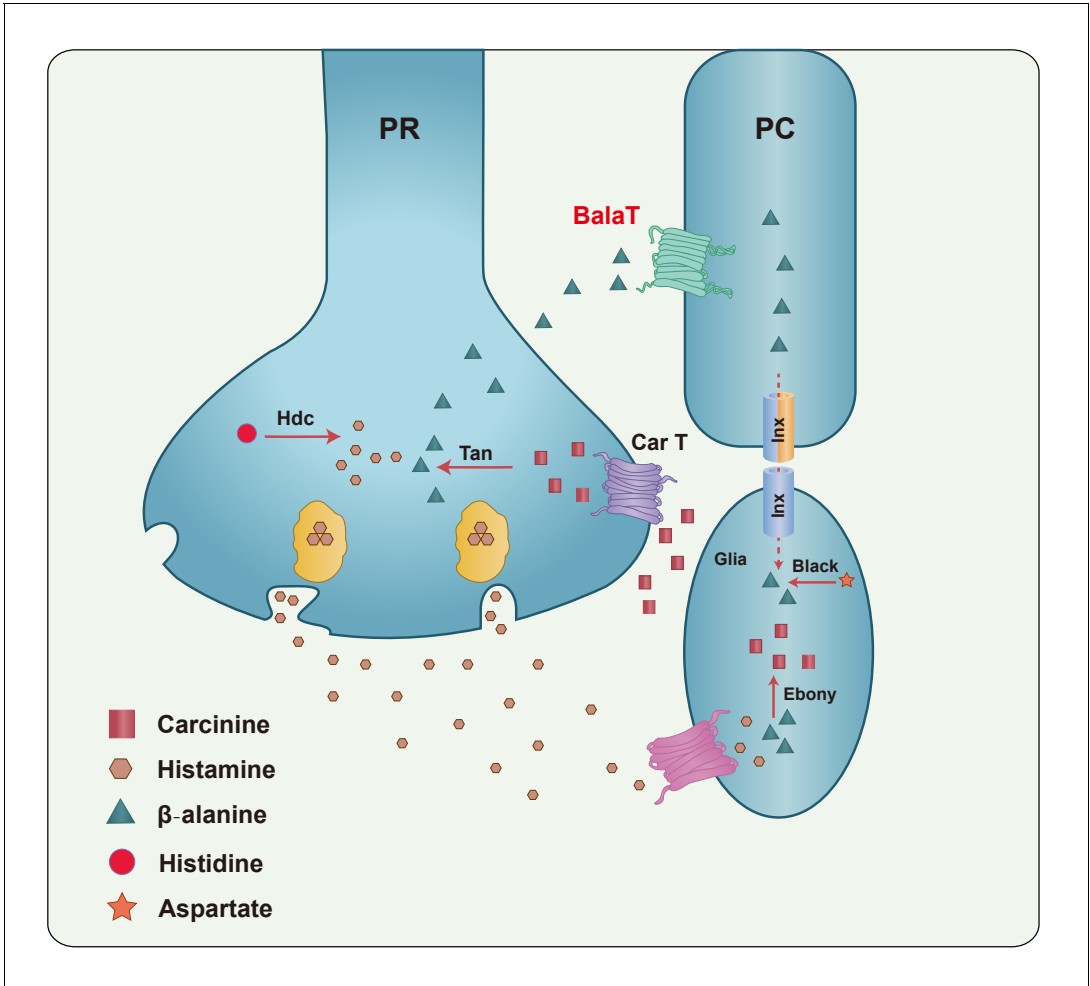

**Figure 6.** A model of the histamine recycling pathway. Histamine is initially synthesized by histidine decarboxylase (Hdc) in photoreceptor cells (PR). Upon light stimulation, PRs release histamine into the synaptic cleft. Released histamine is quickly taken up by an unknown histamine transporter into epithelial glial cells that express Ebony. In these glia, histamine is conjugated to β-alanine, which inactivates histamine and generates carcinine. Carcinine is released into the synaptic cleft and subsequently internalized, via CarT, by the PRs. After carcinine is hydrolyzed to histamine and β-alanine by Tan hydrolase in the PR, histamine is re-packaged into synaptic vesicles, whereas β-alanine is delivered to and subsequently internalized, via the BalaT transporter, by retinal pigment cells (PCs). PCs can store β-alanine or deliver β-alanine to the laminar glia cells through a gap junction network involving Inx1 and Inx3, which are expressed in PCs. In glial cells, β-alanine is conjugated to histamine, and the cycle repeats. Moreover, β-alanine can be synthesized by Black, which is an aspartate decarboxylase that is expressed together with Ebony in optic lobe glia.

## Materials and methods

### Fly stocks

The *pyd1; b$^1$ (su(r) r$^C$; b$^1$)*, *UAS-inx1$^{RNAi}$*, *UAS-inx2$^{RNAi}$*, and *UAS-inx3$^{RNAi}$* flies were provided by the Bloomington *Drosophila* Stock Center. The *nos-Cas9* flies were obtained from Dr. J. Ni at Tsinghua University, Beijing, China. The *cart$^1$* mutant flies were maintained in the lab of Dr. T. Wang (*Xu et al., 2015*).

### Generation of *balat* mutant and knock-in flies

The *balat$^1$* and *balat$^2$* mutations were generated using the Cas9/sgRNA system (*Xu et al., 2015*). Briefly, two pairs of guide RNAs targeting the *balat* locus were designed (sgRNA1: ATATAGTGCGC TATCTTGAG, sgRNA2: GTGTCTACGTGGGACTGAGT, sgRNA3: GAGGCCGGAACACCGGTTTT) and cloned into the *U6b-sgRNA-short* vector. The sgRNA1 and sgRNA2 plasmids were used to generate the *balat$^1$* mutant flies, whereas sgRNA2 and sgRNA3 were used to generate the *balat$^2$* mutant flies. The plasmids were injected into the embryos of *nos-Cas9* flies, and deletions were identified by PCR using the following primers: forward primer 5'-CGCCACCAGTTCCTGGAC-3' and reverse primer 5'-CCAGATGTAAGAGACGCAGTG-3'. The *balat-mcherry* knock-in flies were generated as shown in Figure S3. Briefly, sgDNA sequence was designed and cloned into the *U6b-sgRNA-short* vector. Two fragments from the *balat* locus (−1327 to −38 and +380 to+1660, where + 1 represents the transcription start site) were subcloned into a *pDM19-mcherry* vector such that they were separated by the *3XP3-mcherry* marker. The two plasmids were co-injected into the embryos of *nos-Cas9* flies, and mCherry-positive progeny were identified. By crossing these flies with *hs-Cre* flies, the 3XP3 promoter region was deleted. The *balat-mcherry* flies were finally confirmed by PCR using the following primers: forward primer 5'-AATTAATTAATGCGCACGA-GAGGCC-3' and reverse primer 5'-TTCCACGATGGTGTAGTCCTCGTTG-3'.

### Generation of plasmid constructs and transgenic flies

20 head-enriched potential transporters and the *balat* cDNA sequences were amplified from cDNA clones obtained from DGRC (Drosophila Genomics Resource Center, Bloomington, IN, USA). The mouse GAT3 cDNA sequences were amplified from mouse liver cDNA obtained from mRNA. Their entire CDS sequences were subcloned into the pCDNA3 vector (Invitrogen, Carlsbad, USA) for expression in HEK293T cells and PIB vector (Invitrogen, Carlsbad, USA) for expression in S2 cells. To construct *PninaE-balat*, *Ppdh-balat*, and *PGMR-balat*, the entire coding region of *balat* was amplified from cDNA clones and subcloned into the *pninaE-attB*, *ppdh-attB* and *pGMR-attB* vectors (*Xu et al., 2015*). These constructs were injected into *M(vas-int.Dm) ZH-2A;M(3xP3-RFP.attP) ZH-86Fb* embryos, and transformants were identified on the basis of eye color. The *3xP3-RFP.attP* locus was removed by crossing with *P(Crey)* flies. HEK 293 T cells was obtained from ATCC (RRID:CVCL_0063) and S2 cells was obtained from Drosophila Genomics Resource Center (RRID:CVCL_Z232), which have been tested for contamination of bacteria, yeast, mycoplasma and virus and has been characterized by isozyme and karyotype analysis.

### Electroretinogram recordings

ERGs were recorded as described (*Xu et al., 2015*). Briefly, two glass microelectrodes were filled with Ringer's solution and placed on the surfaces of the compound eye and thorax (one each surface). The source light intensity was ~2000 lux, and the light color was orange (source light was filtered using a FSR-OG550 filter). ERG signals were amplified with a Warner electrometer IE-210, and recorded with a MacLab/4 s A/D converter and the clampelx 10.2 program (Warner Instruments, Hamden, USA).

### β-alanine, histamine, and GABA uptake assay

Alanine, *β*-[3-$^3$H (N)] (30–60 Ci/mM, American radiolabeled chemicals, Saint Louis, USA) uptake was measured as described (*Tomi et al., 2008*). Briefly, pcDNA3/RFP/BalaT or mock-transfected HEK293T cells were cultured in 12-well plates (BD-Falcon) and washed with 1 mL extracellular fluid (ECF) buffer consisting of 120 mM NaCl, 25 mM NaHCO$_3$, 3 mM KCl,1.4 mM CaCl$_2$,1.2 mM MgSO$_4$, 0.4 mM K$_2$HPO$_4$,10 mM D-glucose, and 10 mM Hepes (pH 7.4) at 37°C. Uptake was initiated by

applying 200 µL ECF buffer containing 7400 Bq [³H] β-alanine at 37°C. After 10 min, uptake was terminated by removing the solution, and cells were washed with 1 mL ice-cold ECF buffer. The cells were then solubilized in 1 N NaOH and subsequently neutralized. An aliquot was taken to measure radioactivity and protein content using a liquid scintillation counter and a DC protein assay kit (Bio-rad, USA), respectively, with bovine serum albumin as a standard. Histamine, histamine [ring, Methyl-enes-3H(N)] dihydrochloride, (10–40 Ci/mM, American radiolabeled chemicals, Saint Louis, USA), and GABA, 4-amino-n-[2, 3-³H]butyric acid ([³H]GABA, 30–60 Ci/mmol, PerkinElmer, Waltham, Massachusetts, USA) uptake assays were initiated by applying 200 µL ECF buffer containing 7400 Bq radiolabeled histamine or radiolabeled GABA. The mixture was incubated for 30 min. The continuous procedures are similar to the β-alanine transport assay.

## Immunohistochemistry

Fly head sections (10 µm thick) were prepared from adults that were frozen in OCT medium (Tissue-Tek, Torrance, USA). Immunolabeling was performed on cryosections sections with rabbit anti-β-alanine (1:200, Abcam, USA; RRID:AB_305476), rat anti-RFP (1:200, Chromotek, Germany; RRID:AB_2336064), rabbit anti-PDH (1:200; RRID:AB_2570065) (*Wang et al., 2010*) as primary antibodies. Cells transfected with BalaT-3XFLAG were incubated with mouse monoclonal anti-Flag M2 antibody (1:200, Sigma-Aldrich, USA; RRID:AB_439685) and. Goat anti-rabbit IgG conjugated to Alexa 488 (1:500, Invitrogen, USA; RRID:AB_143165), goat anti-rat IgG conjugated to Alexa 568 (1:500, Invitrogen, USA; RRID:AB_2534121), and goat anti-mouse IgG conjugated to Alexa 647 (1:500, Jackson ImmunoResearch, USA; RRID:AB_2535805) were used as secondary antibodies. Phalloidin conjugated to Alexa 650 (1:500, Thermo scientific, Germany; RRID:AB_2532159) and DAPI (1:1000, Life Technologies, USA; RRID:AB_2307445) was used to indicate photoreceptor cells and nucleus respectively. The images were recorded with a Nikon A1-R confocal microscope.

## The phototaxis assay

Flies were dark adapted for 15 min before phototaxis assay, as described (*Xu et al., 2015*). Phototaxis index was calculated by dividing the total number of flies by the number of flies that walked above the mark. At least three groups of flies were collected for each genotype and three repeats were made for each group. Each group contained ≥20 flies. Results were expressed as the mean of the mean values for the three groups.

## RNA extraction and qPCR

Total RNA was prepared from the heads of 3-day-old flies using Trizol reagent (Invitrogen, Carlsbad, USA), followed by TURBO DNA-free DNase treatment (Ambion, Austin, USA). Total cDNA was synthesized using an iScript cDNA synthesis kit (Bio-Rad Laboratories, USA). iQ SYBR green supermix was used for real-time PCR (Bio-Rad Laboratories, USA). Three different samples were collected from each genotype. The primers used for qPCR were as follows:

*ninaE-fwd*, 5'-ACCTGACCTCGTGCGGTATTG-3'
*ninaE-rev*, 5'-GGAGCGGAGGGACTTGACATT-3'
g*pdh-fwd*, 5'-GCGTCACCTGAAGATCCCATG-3'
*gpdh-rev*, 5'-CTTGCCATACTTCTTGTCCGT-3'
*pdh-fwd*, 5'-GCTTGGCGAACGAAAAGTACT-3'
*pdh-rev*, 5'-GTCACTCGTTTCCGGGAAGAT-3'
*balat-fwd*, 5'- AGACATCACACTGCTGCTCTAC -3'
*balat-rev*, 5'- CCTCCTCCAAAGTCTGTGGAAG -3'

## Liquid chromatography–mass spectrometry (LC-MS)

The Dionex Ultimate 3000 UPLC system was coupled to a TSQ Quantiva Ultra triple-quadrupole mass spectrometer (Thermo Fisher, CA), equipped with a heated electrospray ionization (HESI) probe in negative ion mode. Extracts were separated by a Fusion-RP C18 column (2 × 100 mm, 2.5 µm, phenomenex). Data acquired in selected reaction monitoring (SRM) for histamine, carcinine, and β-alanine with transitions of 112/95.2, 183/95, and 90/72, respectively. Both precursor and fragment ions were collected with resolution of 0.7 FWHM. The source parameters are as follows: spray voltage: 3000 V; ion transfer tube temperature: 350°C; vaporizer temperature: 300°C; sheath gas flow

rate: 40 Arb; auxiliary gas flow rate: 20 Arb; CID gas: 2.0 mTorr. Data analysis and quantification were performed using the software Xcalibur 3.0.63 (Thermo Fisher, CA). Each sample contained 50 Drosophila heads, and the mean values from five samples were calculated.

## Acknowledgements

We thank the Bloomington Stock Center, Tsinghua fly center, and the Developmental Studies Hybridoma Bank for stocks and reagents. We thank Y Wang and X Liu for assistance with fly injections. We are tremendously thankful for support provided by the Metabolomics Facility at the Technology Center for Protein Sciences, Tsinghua University. We thank Ms. Ping Wu from the Imaging Facility of the National Center for Protein Science in Beijing for assistance with microscopy. We thank Dr. D O'Keefe for comments on the manuscript.

## Additional information

### Funding

| Funder | Grant reference number | Author |
|---|---|---|
| Chinese Ministry of Science | 2014CB849700 | Tao Wang |
| National Natural Science Foundation of China | 81670891 | Tao Wang |

The funders had no role in study design, data collection and interpretation, or the decision to submit the work for publication.

### Author contributions

YH, Conceptualization, Funding acquisition, Validation, Writing—original draft, Project administration, Writing—review and editing; LX, Conceptualization, Data curation, Validation, Investigation, Visualization, Methodology, Writing—original draft, Writing—review and editing; YX, Conceptualization, Validation, Investigation, Visualization, Methodology, Writing—original draft, Writing—review and editing; TT, Conceptualization, Investigation, Methodology; TW, Investigation, Methodology

### Author ORCIDs

Yongchao Han, http://orcid.org/0000-0003-1965-0161
Liangyao Xiong, http://orcid.org/0000-0002-5205-8648
Tao Wang, http://orcid.org/0000-0002-2395-3483

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
