## [Decision Letter]

[Editors’ note: a previous version of this study was rejected after peer review, but the authors submitted for reconsideration. The first decision letter after peer review is shown below.]

Thank you for submitting your work entitled "The β-alanine transporter BalaT is required for visual neurotransmission in *Drosophila*" for consideration by *eLife*. Your article has been reviewed by three peer reviewers, and the evaluation has been overseen by a Reviewing Editor and a Senior Editor. The following individuals involved in review of your submission have agreed to reveal their identity: Hong-Sheng Li (Reviewer #2).

Our decision has been reached after consultation between the reviewers. Based on these discussions and the individual reviews below, we regret to inform you that although the work is of interest, it will not be considered further for publication in *eLife* as the findings at this stage are too preliminary. However, if you can address the comments raised by the reviewers we are willing to re-review a substantially edited version that address the reviewers concern.

The following are important issues to address:

The authors proposed that β-Ala diffuses between retinal and lamina through gap junctions and provided RNAi evidence that either Inx1 or Inx3 forms the junctions. Although this hypothesis represents the most direct model of β-Ala trafficking, EM studies from both *Drosophila* and Musca didn't reveal any gap junction between retinal pigment cells and laminar fenestrated glia.

We know from research reported from the Hovemann lab that epithelial glia express Black, which synthesizes β-alanine from aspartate (Ziegler 2013, J Comp Neurol 521: 1207-1224). In the same paper the coexpression of Ebony in the epithelial glia was also demonstrated. This indicates that the machinery to produce β-alanine and carcinine is exactly present in the site involved in β-alanyl-histamine synthesis and probably should be sufficient to maintain the histamine recycling. It is strange that the authors choose not to report the findings of that study, since they bear critically on this submission.

The authors' interests seem to be one-sided and focus only on β-alanine transport, as shown in Figure 1. As a standard in Figure 1 is presented the function of a transporter that is not specific for β-alanine transport e.g. GAT3. In Figure 1 the tritiated β-alanine uptake into S2 cells is similar for that for CG3790 and for the GABA transporter GAT3. Why have the authors not checked whether CG3790 is a GABA transporter in *Drosophila*? Maybe CG3790 also acts as a transporter for other biologically important and structurally similar molecules, such as glycine, taurine or even glutamate. Transport by the product of CG3790 of some other substances engaged in phototransmission, could explain the ERG abnormalities observed after deleting this gene. Why did the authors not test histamine as a substrate for CG3790 product?

Reviewer #1:

Wang et al. present a straightforward manuscript on the characterization of a novel β-alanine transporter with a specific defect in visual transmission in *Drosophila*. This is an interesting system to study neurotransmitter recycling through support cells, even though the neurotransmitter system itself (histamine) is quite special. The manuscript presents the novel and interesting finding, namely that in addition to glia cells another cell type, the interommatidial pigment cells, have a special requirement in the process: histamine is generated through 'splitting' of histamine and β-alanine from carcinine in photoreceptor terminals; β-alanine is apparently then taken up by the pigment cells through the newly described BalaT transporter for delivery into glia cells, where histamine and β-alanine get conjugated again for delivery back into photoreceptor terminals.

The strength of the manuscript is in the clear straight-forward discovery of the balat transporter using a β-alanine uptake assay and demonstration of its function in pigment cells using cell-specific rescue. The link to glia is a bit weaker: RNAi experiments in two innexins that are supposed to be the delivery route of β-alanine back into the glia are not controlled and not fully convincing. This and a few minor gaps in the story can be fixed in a reasonable amount of time. These fixes should include a better characterization of the *balat-mcherry*, quantification of ERGs and better immunohistochemistry.

The key experiment I would like to see regards the innexin link: if innexin mutants really are the link between pigments cells and glia, then β-alanine should accumulate in the pigment cells in the innexin double RNAi knock-down (which also needs some RNAi controls). With inclusion of these revisions, I am in support of the manuscript.

Reviewer #2:

The *Drosophila* visual system is a great model in the study of neurotransmitter homeostasis, in which the primary visual transmitter, histamine, is recycled through a glial pathway that involves two essential enzymes (Ebony and Tan) and multiple steps of transmembrane transport. Previous genetic and immunostaining data have suggested the involvement of retinal pigment cells in the storage/transport of β-Ala, a histamine-inactivating metabolite during this transmitter recycling. Here Wang et al. identified a β-Ala transporter in the pigment cells and demonstrated its importance to the visual transmission at both electrophysiological and behavioral levels. This is merely the second membrane transporter identified in the histamine recycling pathway, and the work provides direct evidence for the involvement of retinal pigment cells.

My only concern is about the step of β-Ala transport from pigment cells to laminal glia, where β-Ala is conjugated to histamine to form the inactive metabolite carcinine. The authors proposed that β-Ala diffuses between retinal and lamina through gap junctions and provided RNAi evidence that either Inx1 or Inx3 forms the junctions. Although this hypothesis represents the most direct model of β-Ala trafficking, EM studies from both *Drosophila* and Musca didn't reveal any gap junction between retinal pigment cells and laminar fenestrated glia. It is thus important to discuss on alternative, Inx-independent trafficking mechanisms. The ERG defects of Inx1+Inx3 knockdown flies could have different interpretations. For instances, they could be required for the development of synapses between retinal and laminar neurons (see Curtin et al., 2002, J. Cell Sci.), or for the β-Ala transport into core cells for storage.

Reviewer #3:

The authors report a new β-alanine transporter, encoded by CG3790, which they named BalaT, and which is present in the retinal pigment cells of the *Drosophila* visual system. The results indicate that BalaT is critical for β-alanine recycling together with the action of innexins. The authors demonstrate that lacking BalaT disrupts the normal ERG and phototaxis. The results are consistent with the authors' theory but they run contrary to current knowledge about β-alanine and its sources in the *Drosophila* head. We know from research reported from the Hovemann lab that epithelial glia express Black, which synthesizes β-alanine from aspartate (Ziegler 2013, J Comp Neurol 521: 1207-1224). In the same paper the coexpression of Ebony in the epithelial glia was also demonstrated. This indicates that the machinery to produce β-alanine and carcinine is exactly present in the site involved in β-alanyl-histamine synthesis and probably should be sufficient to maintain the histamine recycling. It is strange that the authors choose not to report the findings of that study, since they bear critically on this submission. What do the authors suppose is the reason that the long way to transport β-alanine into the epithelial glia, as suggested in their manuscript, would offer adaptive advantage and be critical for histamine recycling? It seems obvious that the more energetically demanding process should be outcompeted by the simpler one. The authors have forgotten to mention, or possibly have overlooked the Ziegler et al. (2013) paper in their manuscript, and in this way have avoided discussing the strongest arguments against their proposed β-alanine recycling pathway. From the work of others (Borycz et al. 2012, J Exp Biol 215: 1399-1411; Ziegler 2013, J Comp Neurol 521: 1207-1224; Rawls 2006, Genetics 172: 1665-1674) it is known that the metabolism of uracil produces β-alanine, so it is a substance present in all living cells. These biochemical processes are not the only ones and there are also reports of alternative pathways for β-alanine synthesis. Borycz et al. (2012) reported more than 10 ng β-alanine/head in the wild type, which is 5 times the histamine head content in these flies. The process of histamine conjugation with β-alanine, mediated by Ebony in the epithelial glia, requires roughly equal amounts of both compounds, which indicates that head β-alanine greatly exceeds the requirements for Ebony function. The abundance of β-alanine may in fact indicate that there is no need to recycle this substance. For these reasons the controversy with the Ziegler at al. (2012) paper should be resolved, and could only be considered in a resubmitted version.

The authors' interests seem to be one-sided and focus only on β-alanine transport, as shown in Figure 1. As a standard in Figure 1 is presented the function of a transporter that is not specific for β-alanine transport e.g. GAT3. In Figure 1 the tritiated β-alanine uptake into S2 cells is similar for that for CG3790 and for the GABA transporter GAT3. Why have the authors not checked whether CG3790 is a GABA transporter in *Drosophila*? Maybe CG3790 also acts as a transporter for other biologically important and structurally similar molecules, such as glycine, taurine or even glutamate.

Transport by the product of CG3790 of some other substances engaged in phototransmission, could explain the ERG abnormalities observed after deleting this gene. Why did the authors not test histamine as a substrate for CG3790 product? The histamine transporter has yet to be discovered and histamine is the neurotransmitter of major importance in the fly's visual system.

Additionally, use of human embryonic kidney cells (HEK293T) to study the invertebrate transporter function may be methodologically a wrong decision. The concentrations of some ions and molecules may be different in insect and human cells, so that the observed effect may be irrelevant to the results that could be seen in the appropriate cells/tissues.

The metabolite β-alanine is structurally similar to L-alanine or D-alanine as well as to GABA and glycine. These substances should all also be analysed when assessing the uptake (Figure 1) and also considered as a source of noise observed in immunolabeling (Figure 5).

Finally, the legends for the figures are scarce and abbreviations are often not explained, neither in the legend nor in the text. For example, to what does RFP in Figure 1 refer?

The results connected to the role of innexins are interesting and may indicate their role in balancing the neuronal voltage gradients but again their link to β-alanine may be misleading. Altogether, before it could be reconsidered, the paper requires substantial revision and addressing all important questions related to the previous β-alanine findings. The only link to β-alanine transport provided by the authors are the data shown in Figure 1, because the ERGs and phototaxis fail to differentiate between β-alanine and other factors involved in the process.

To report a protein as a transporter requires more detailed analysis of the transporter kinetics than the data shown in the manuscript.

In addition, the flies used in the study all contained a background of *w1118*, which is itself an ABC-transporter-deficient mutant, and this may be a factor adding to the final results.

In the list of references for *eLife*, the authors should be listed in alphabetical order and not chronological one. This omission suggests a lack of care in preparation of the manuscript for submission. Overall I cannot recommend this submission for publication in its present form.

---

## [Author Response]

[Editors’ note: the author responses to the first round of peer review follow.]

Reviewer #1:

[…] The strength of the manuscript is in the clear straight-forward discovery of the balat transporter using a β-alanine uptake assay and demonstration of its function in pigment cells using cell-specific rescue. The link to glia is a bit weaker: RNAi experiments in two innexins that are supposed to be the delivery route of β-alanine back into the glia are not controlled and not fully convincing. This and a few minor gaps in the story can be fixed in a reasonable amount of time. These fixes should include a better characterization of the balat-mcherry, quantification of ERGs and better immunohistochemistry.

We improved the immunostaining assays in *balat-mcherry* flies, and now show these results in Figure 3. We also quantified the ERG OFF transients (data are shown in Figure 2).

The key experiment I would like to see regards the innexin link: if innexin mutants really are the link between pigments cells and glia, then β-alanine should accumulate in the pigment cells in the innexin double RNAi knock-down (which also needs some RNAi controls). With inclusion of these revisions, I am in support of the manuscript.

We performed the experiments as suggested. We performed β-alanine immunostaining to examine the in vivo levels of β-alanine in heads of control (knocking down of CG9962 and CG2198, which do not express in compound eyes), and Inx1/Inx3 double knockdown flies. These results have been added to Figure 4, and described in the Results section (subsection “β-alanine levels and distributions are altered in *balat* mutant flies”, last paragraph). In Inx1/Inx3 double knockdown flies, β-alanine levels were slightly increased and largely accumulated in the retinal layer compared with controls, which confirmed our hypothesis that Inx1/Inx3 functions to transport β-alanine back to glia cells.

Reviewer #2:

[…] My only concern is about the step of β-Ala transport from pigment cells to laminal glia, where β-Ala is conjugated to histamine to form the inactive metabolite carcinine. The authors proposed that β-Ala diffuses between retinal and lamina through gap junctions and provided RNAi evidence that either Inx1 or Inx3 forms the junctions. Although this hypothesis represents the most direct model of β-Ala trafficking, EM studies from both Drosophila and Musca didn't reveal any gap junction between retinal pigment cells and laminar fenestrated glia. It is thus important to discuss on alternative, Inx-independent trafficking mechanisms. The ERG defects of Inx1+Inx3 knockdown flies could have different interpretations. For instances, they could be required for the development of synapses between retinal and laminar neurons (see Curtin et al., 2002, J. Cell Sci.), or for the β-Ala transport into core cells for storage.

We have analyzed β-alanine levels and performed β-alanine immunostaining of inx1/inx3 double knock-down mutants (data is shown in Figure 4). Β-alanine largely accumulated in the retinal layer. This may represent direct evidence that Inx1/Inx3 functions to transport β-alanine back to glia cells. However, we have been unable to detect gap junctions between these cells at the EM level. Therefore, we discuss the possibilities that developmental roles of Inx may be an alternative explanation (Discussion, fifth paragraph).

Reviewer #3:

The authors report a new β-alanine transporter, encoded by CG3790, which they named BalaT, and which is present in the retinal pigment cells of the Drosophila visual system. The results indicate that BalaT is critical for β-alanine recycling together with the action of innexins. The authors demonstrate that lacking BalaT disrupts the normal ERG and phototaxis. The results are consistent with the authors' theory but they run contrary to current knowledge about β-alanine and its sources in the Drosophila head. We know from research reported from the Hovemann lab that epithelial glia express Black, which synthesizes β-alanine from aspartate (Ziegler 2013, J Comp Neurol 521: 1207-1224). In the same paper the coexpression of Ebony in the epithelial glia was also demonstrated. This indicates that the machinery to produce β-alanine and carcinine is exactly present in the site involved in β-alanyl-histamine synthesis and probably should be sufficient to maintain the histamine recycling. It is strange that the authors choose not to report the findings of that study, since they bear critically on this submission. What do the authors suppose is the reason that the long way to transport β-alanine into the epithelial glia, as suggested in their manuscript, would offer adaptive advantage and be critical for histamine recycling? It seems obvious that the more energetically demanding process should be outcompeted by the simpler one. The authors have forgotten to mention, or possibly have overlooked the Ziegler et al. (2013) paper in their manuscript, and in this way have avoided discussing the strongest arguments against their proposed β-alanine recycling pathway. From the work of others (Borycz et al. 2012, J Exp Biol 215: 1399-1411; Ziegler 2013, J Comp Neurol 521: 1207-1224; Rawls 2006, Genetics 172: 1665-1674) it is known that the metabolism of uracil produces β-alanine, so it is a substance present in all living cells. These biochemical processes are not the only ones and there are also reports of alternative pathways for β-alanine synthesis. Borycz et al. (2012) reported more than 10 ng β-alanine/head in the wild type, which is 5 times the histamine head content in these flies. The process of histamine conjugation with β-alanine, mediated by Ebony in the epithelial glia, requires roughly equal amounts of both compounds, which indicates that head β-alanine greatly exceeds the requirements for Ebony function. The abundance of β-alanine may in fact indicate that there is no need to recycle this substance. For these reasons the controversy with the Ziegler at al. (2012) paper should be resolved, and could only be considered in a resubmitted version.

It is our mistake that we did not discuss in detail the relationship between de novo synthesis and recycle of β-alanine. To remedy this, we have added two paragraphs to the Discussion (second and third paragraphs).

In wild type flies, β-alanine levels in the head are 5 times higher than histamine. Even in pyd and black double mutant flies, β-alanine levels are still higher than histamine. Black is expressed in Ebony-expressing lamina glia, and serves as the major enzyme in the local generation of β-alanine. The pyrimidine catabolism pathway is an alternative pathway for β-alanine synthesis. However, the *pyd* and *black* double mutant flies do not completely lose the ability to synthesize β-alanine, which may be due to the uptake of exogenous β-alanine. This low level of β-alanine is enough to sustain visual transduction. If β-alanine does not need to be recycled, the phenotype should be the same as seen in ebony mutants. Therefore, the normal visual response in black and *pyd,black* double mutants indicates that the β-alanine shuttle system indeed exists in the visual system. In *balat* mutant flies, levels of β-alanine in the head are reduce as well, but remain a few times higher than histamine. However, *balat* mutant flies display obvious ERG phenotypes, as they lack “on” and “off” transients. If the abundance of β-alanine indicates that there is no need to recycle this substance, we should not observe ERG phenotypes and β-alanine decreases in *balat* mutant fly heads. We could explain this by the β-alanine distribution changes which indicates that the β-alanine trafficking is blocked.

It has been demonstrated in many cases that de novo synthesis and the recycling of metabolites are both critical to maintain their threshold levels. For example, histamine is easily generated by Hdc in photoreceptor cells. However, the recycling of histamine to photoreceptor cells is also important to sustain visual transduction. Loss of either pathway disrupted visual transduction. However, loss of visual transduction caused by mutations in *Hdc* (which disrupts de novo synthesis) can be fully restored by feeding histamine. The same situation occurred in pyd and black double mutant flies, in which exogenous β-alanine is enough to keep β-alanine levels over histamine. In contrast, feeding histamine did not restore visual transduction in ebony mutants (which affect the recycle pathway). This situation is similar to balat-dependent β-alanine trafficking, which is required for synaptic transmission even though local de novo synthesis takes place in glia cells.

The authors' interests seem to be one-sided and focus only on β-alanine transport, as shown in Figure 1. As a standard in Figure 1 is presented the function of a transporter that is not specific for β-alanine transport e.g. GAT3. In Figure 1 the tritiated β-alanine uptake into S2 cells is similar for that for CG3790 and for the GABA transporter GAT3. Why have the authors not checked whether CG3790 is a GABA transporter in Drosophila? Maybe CG3790 also acts as a transporter for other biologically important and structurally similar molecules, such as glycine, taurine or even glutamate.

We have assessed the transporter specificity of CG3790. CG3790 did not transport Histamine or GABA. We have added these data to figures (Figure 1—figure supplement 2) and to the Results section (subsection “CG3790 transports β-alanine in vitro”, second paragraph).

Transport by the product of CG3790 of some other substances engaged in phototransmission, could explain the ERG abnormalities observed after deleting this gene. Why did the authors not test histamine as a substrate for CG3790 product? The histamine transporter has yet to be discovered and histamine is the neurotransmitter of major importance in the fly's visual system.

We preformed histamine uptake assays, which revealed that CG3790 does not exhibit histamine uptake activity. As a control, the human Organic Cation Transporter (OCT2), which is known to take up histamine, exhibited high levels of histamine transport activity (Figure 1—figure supplement 2 and Results subsection “CG3790 transports β-alanine in vitro”, second paragraph”).

Additionally, use of human embryonic kidney cells (HEK293T) to study the invertebrate transporter function may be methodologically a wrong decision. The concentrations of some ions and molecules may be different in insect and human cells, so that the observed effect may be irrelevant to the results that could be seen in the appropriate cells/tissues.

For ease of manipulation we used HEK293T cells to screen for putative fly β-alanine transporters. We then performed all CG3790 transport experiments in S2 cells (Figure 1 and Figure 1—figure supplement 2).

The metabolite β-alanine is structurally similar to L-alanine or D-alanine as well as to GABA and glycine. These substances should all also be analysed when assessing the uptake (Figure 1) and also considered as a source of noise observed in immunolabeling (Figure 5).

We asked whether CG3790 is able to transport GABA. In contrast to mGAT3, which exhibited high levels of GABA transporting activity, CG3790 did not show any GABA transporting activity when expressed in S2 cells (subsection “CG3790 transports β-alanine in vitro”, second paragraph).

Finally, the legends for the figures are scarce and abbreviations are often not explained, neither in the legend nor in the text. For example, to what does RFP in Figure 1 refer?

We have reviewed the figure legends and fixed the abbreviations. For example, we have changed “RFP” to “red fluorescent protein (RFP)” as a negative control in Figure 1 figure legend.

The results connected to the role of innexins are interesting and may indicate their role in balancing the neuronal voltage gradients but again their link to β-alanine may be misleading. Altogether, before it could be reconsidered, the paper requires substantial revision and addressing all important questions related to the previous β-alanine findings. The only link to β-alanine transport provided by the authors are the data shown in Figure 1, because the ERGs and phototaxis fail to differentiate between β-alanine and other factors involved in the process.

We performed β-alanine immunostaining to examine the in vivo levels of β-alanine in heads of Inx1/Inx3 double knockdown flies. These results have been added to Figure 4, and described in the Results section (subsection “β-alanine levels and distributions are altered in *balat* mutant flies”, last paragraph”). In Inx1/Inx3 double knockdown flies, β-alanine levels were slightly increased and largely accumulated in the retinal layer compared with controls, which confirmed our hypothesis that Inx1/Inx3 functions to transport β-alanine back to glia cells.

To report a protein as a transporter requires more detailed analysis of the transporter kinetics than the data shown in the manuscript.

We have strong evidence to show that Balat is specific β-alanine transporter. Although kinetic data would be a nice addition to the manuscript, we do not think these data would dramatically improve the manuscript.

In addition, the flies used in the study all contained a background of w1118, which is itself an ABC-transporter-deficient mutant, and this may be a factor adding to the final results.

To rule out the influence of the *white* gene, we crossed the *balat* mutation into the *w^+^* background. The “on” and “off” transients were still absent. We have added these data to Figure 2, and to the Results section (subsection “BalaT is required for visual synaptic transmission”, second paragraph).

In the list of references for eLife, the authors should be listed in alphabetical order and not chronological one. This omission suggests a lack of care in preparation of the manuscript for submission. Overall I cannot recommend this submission for publication in its present form.

We have listed the references in alphabetical order.